# Effect of High-Sucrose Diet on the Occurrence and Progression of Diabetic Retinopathy and Dietary Modification Strategies

**DOI:** 10.3390/nu16091393

**Published:** 2024-05-05

**Authors:** Chen Yang, Yifei Yu, Jianhong An

**Affiliations:** 1State Key Laboratory of Ophthalmology, Optometry and Vision Science, Wenzhou Medical University, Wenzhou 325027, China; yangchen_0416@163.com; 2Oujiang Laboratory, Zhejiang Lab for Regenerative Medicine, Vision and Brain Health, Wenzhou 325101, China; 3Key Laboratory of Precision Nutrition and Food Quality, Department of Nutrition and Health, China Agricultural University, Beijing 100193, China

**Keywords:** high-sucrose diet, diabetic retinopathy, hyperglycemia, dietary modification strategies

## Abstract

As the most serious of the many worse new pathological changes caused by diabetes, there are many risk factors for the occurrence and development of diabetic retinopathy (DR). They mainly include hyperglycemia, hypertension, hyperlipidemia and so on. Among them, hyperglycemia is the most critical cause, and plays a vital role in the pathological changes of DR. High-sucrose diets (HSDs) lead to elevated blood glucose levels in vivo, which, through oxidative stress, inflammation, the production of advanced glycation end products (AGEs) and vascular endothelial growth factor (VEGF), cause plenty of pathological damages to the retina and ultimately bring about loss of vision. The existing therapies for DR primarily target the terminal stage of the disease, when irreversible visual impairment has appeared. Therefore, early prevention is particularly critical. The early prevention of DR-related vision loss requires adjustments to dietary habits, mainly by reducing sugar intake. This article primarily discusses the risk factors, pathophysiological processes and molecular mechanisms associated with the development of DR caused by HSDs. It aims to raise awareness of the crucial role of diet in the occurrence and progression of DR, promote timely changes in dietary habits, prevent vision loss and improve the quality of life. The aim is to make people aware of the importance of diet in the occurrence and progression of DR. According to the dietary modification strategies that we give, patients can change their poor eating habits in a timely manner to avoid theoretically avoidable retinopathy and obtain an excellent prognosis.

## 1. Introduction

Carbohydrates are one of the three primary macronutrients essential for human energy production and are an important source of energy for maintaining normal physiological activities and functions [1]. However, long-term excessive intake of carbohydrates, which is a high-sucrose diet (HSD), especially foods containing high amounts of glucose and fructose, can lead to increased sugar intake and cause metabolic disorder, cardiovascular abnormalities [2], neurological disturbances [1,3] and inflammation [4,5,6]. This can manifest as hyperglycemia [7], obesity [8,9,10], insulin resistance [8], hypertension [11], impaired cardiac metabolic function [12], mood and behavioral disorders, impaired working memory [13], food addiction [1,14], intestinal infections and microbial dysbiosis [15,16]. In addition, HSDs can also contribute to psychopathology [17]. An HSD in women of childbearing age and during pregnancy can impact the memory processes of offspring [18] and may contribute to a predisposition to developing mental disorders in early life or adulthood.

As the most severe of the many worse new pathological microvascular changes caused by diabetes, diabetic retinopathy (DR) is the leading pathogeny of preventable vision loss in the young, especially working-age people worldwide [19]. The International Diabetes Federation (IDF) estimates that, by 2045, the global population aged 20–79 with diabetes will rise to 780 million [20]. It is estimated that 160 million adults will become DR sufferers [21]. Clinically, the confirmation of DR is primarily on the basis of the abnormal appearance of retinal vessels [22,23] (see Figure 1). There are two central disease stages. The first is non-proliferative diabetic retinopathy (NPDR), which is characterized by progressive retinal microvascular lesions. Then, it progresses to the proliferative diabetic retinopathy (PDR) stage, which is characterized by neovascularization [24,25]. NPDR involves retinal changes such as hemorrhages, microaneurysms and hard exudates, with the patients usually being asymptomatic. As NPDR progresses and retinal ischemia occurs, leading to neovascularization of the retina, it advances to PDR, and patients may experience severe visual impairment [24,25]. Additionally, diabetic macular oedema (DMO) refers to the thickening of the posterior pole of the retina and can occur at any stage [25]. Due to the late onset of clinical symptoms, early histological changes are challenging to detect during clinical examinations, and, once detected, irreversible vision loss often occurs. Beyond vision impairment, the degree of retinal damage in diabetic sufferers is significantly related to future risks of cerebrovascular accidents, myocardial infarction and mortality [26]. Therefore, for patients with DR, earlier intervention and timely diagnosis and treatments are necessary for patients with DR to reduce the potential risk of vision loss [27]. Controlling hyperglycemia is the most crucial preventive measure [28].

The risk factors that primarily contribute to the occurrence and progression of DR include hyperglycemia, hypertension, hyperlipidemia, diabetes duration and so on. Among these factors, hyperglycemia is the critical factor that can trigger all the related abnormalities. Although there are currently many medical treatment options targeted toward DR, they are only applicable to the terminal stage of the disease and often come with severe adverse reactions [29], posing a significant economic burden on patients and the global public healthcare system. The improper control and management of DR can lead to late-stage DR [30], which may result in blindness and increase the burden of DR disease [31]. Studies have shown that maintaining normal blood glucose levels can effectively delay the occurrence and progression of DR [32]. A high-sucrose diet and the resultant hyperglycemia exacerbate the occurrence and progression of DR, leading to vision impairment in patients.

## 2. Risk Factors for the Occurrence and Development of DR

The primary connected factor is blood glucose levels [33]. An HSD significantly elevates blood glucose levels [34].

### 2.1. Hyperglycemia

Hyperglycemia significantly affects DR [35] and is the dominating pathogeny of DR progression [36,37]. Glycated hemoglobin A1c (HbA1c) is a marker applied to measure blood glucose control [38], and its blood level represents the average blood glucose concentration over the past 120 days [39]. Strictly controlling HbA1c levels below 7% can lower the risk of DR occurrence and development [22]. Every 1% reduction in HbA1c reduces the risk of retinopathy and blindness by 40% and 15%, respectively [40]. It is a remarkable fact that earlier blood glucose control is better because long-term exposure to high blood glucose conditions can cause irreversible retinal damage even after blood glucose control is regained [41].

### 2.2. Hypertension

Hypertension is defined as a blood pressure reading of 130/80 mmHg or higher [42]. Hypertension stands as an independent risk factor for the onset of retinopathy in individuals diagnosed with type 2 diabetes mellitus [43]. Strictly controlling the pressure of blood can reduce the risk of DR and vision loss [44], especially when maintaining HbA1c levels around 7% simultaneously [45].

### 2.3. Hyperlipidemia

Hyperlipidemia refers to an elevation in circulating levels of low-density lipoprotein cholesterol (LDL-C) and very-low-density lipoprotein cholesterol (VLDL-C), accompanied by decreased levels of high-density lipoprotein cholesterol (HDL-C), which holds protective attributes [46]. Lipid-lowering agents, including statins, employed to manage hypercholesterolemia, demonstrate a reduction in the risk of DR incidence [47,48]. Intensifying blood glucose control in combination with treating hyperlipidemia can slow down the progression of DR [49].

### 2.4. Duration of Diabetes Mellitus

The earlier the patients have diabetes, the greater the risk of DR [50]. This phenomenon arises from the prolonged duration of diabetes, signifying a persistent impact of hyperglycemia on the body, and, consequently, an enduring assault on the retina. Research indicates that, with a diabetic duration exceeding 30 years, the prevalence of retinopathy can soar to 63% [51].

### 2.5. Other Risk Factors

In addition to the aforementioned primary factors, ethnic origin, pregnancy and puberty are also additional risk factors for DR. Individuals of South Asian or African descent exhibit a higher prevalence of diabetic DR compared to Caucasians [52]. Pregnancy, particularly during mid-term gestation, exerts detrimental effects on retinal vasculature due to fluctuations in estrogen levels and increased blood volume [53]. Suboptimal glycemic control during puberty correlates closely with the progression of diabetic retinopathy [54], underscoring the significance of blood glucose levels in its advancement.

## 3. Pathophysiology of DR

DR primarily involves retinal microangiopathy, with histological alterations occurring before the emergence of various obvious clinical symptoms. Persistent hyperglycemia assumes a pivotal role in the pathological advancement of DR by instigating and maintaining various other factors that collectively influence the development of DR.

DR encompasses a classification system comprising two fundamental clinical stages: NPDR and PDR. NPDR signifies the initial phase of DR, whereas PDR confers an escalated propensity toward vision loss relative to individuals with NPDR [55]. The primary pathology of DR involves microvascular changes in the retinal capillaries, comprising pericytes, basement membranes and endothelial cells. Pericytes possess contractile properties, maintaining capillary tone, controlling capillary diameter, regulating blood flow in the capillaries and preserving their stability [56]. Endothelial cells form tight junctions, creating an internal barrier to prevent substances from leaking out of the blood vessels [57]. During the initial phases of DR progression, elevated blood glucose levels instigate pericyte death, resulting in capillary acellular areas and the development of localized or diffuse microaneurysms due to capillary rarefaction [58]. High blood glucose damages endothelial cells, compromising the integrity of the internal barrier and causing vascular leakage [59]. Increased thickness of the basement membrane induces luminal constriction within the vasculature and vascular stiffness, thereby promoting vascular rigidity and impeding the binding efficacy of growth factors. The loss of these two cell types leads to capillary leakage and occlusion, resulting in non-perfused areas. As the disease progresses, arteriolar involvement occurs, and dilated retinal arterioles accelerate the clinical manifestation of diabetic retinopathy, including edema and hemorrhage [60]. The non-perfused areas further expand, causing capillary dilation and venous beading formation, which are known as intraretinal microvascular abnormalities [40]. NPDR is characterized by various manifestations, including small dilations of blood vessels in the microvascular network, blood leakage, irregularities in the veins of the retinal vasculature, a reduction in function capillaries and intraretinal microvascular abnormalities [61]. Notably, retinal arteriolar dilatation potentially serves as an early subclinical marker of microvascular dysfunction [22].

Extensive death of retinal microvascular cells leads to retinal hypoxia, triggering the upregulation of growth factors [62]. Various growth factors act synergistically, resulting in the formation of neovascularization [63], marking the transition to the proliferative stage of DR. Neovascularization continues to grow and forms fibrous tissue membranes. Subsequently, these fibrous tissues adhere to the vitreous. Contraction of the vitreous can cause hemorrhage, and the contracted fibrous tissue leads to retinal traction and detachment, ultimately resulting in vision loss [64].

Additionally, during the initial phases of the ailment, there is an associated impairment of both retinal ganglion and glial cells [65]. Neurologic and glial dysfunction occur concurrently with vascular abnormalities and generally precede obvious microvascular damage [66].

## 4. Molecular Mechanisms of HSD-Induced Development of DR

Sugar (sucrose) includes fructose and glucose [67]. An HSD can result in elevated levels of glucose in the bloodstream. Elevated blood glucose levels are pivotal in the pathogenesis of vascular complications associated with diabetes. DR represents the predominant complication of the microvasculature. Extensive research has demonstrated that hyperglycemia, especially long-term sustained high blood glucose levels, serves as a central factor in the occurrence and progression of DR, inducing a variety of biochemical abnormalities [68,69] (Figure 2).

### 4.1. Oxidative Stress

Under optimal physiological circumstances, the body maintains a meticulous oxidative–reductive equilibrium. However, disruption ensues when there is a disproportionate interplay between the generation and elimination of free radicals, thereby perturbing this dynamic equilibrium and instigating an upsurge in free radical production [70]. Consequently, oxidative stress manifests itself [71]. Given the retina’s unique characteristics, characterized by protracted light exposure, heightened oxygen consumption and reliance on glucose oxidation, it becomes highly susceptible to oxidative stress [72]. The confluence of hyperglycemia precipitates the accrual of reactive oxygen species (ROS) within the retina, thereby subjecting retinal and capillary cells to oxidative stress [59,73,74]. Subsequently, diverse molecular mechanisms are initiated to cause oxidative stress on retinal well-being, including the activation of the protein kinase C (PKC) pathway, formation of advanced glycation end products (AGEs), initiation of the polyol pathway and facilitation of the hexosamine pathway [75], as well as the induction of inflammatory cascades [71,76,77].

### 4.2. Inflammation

Hyperglycemia disrupts the balance between pro-inflammatory and anti-inflammatory responses maintained by microglia, leading to a state of inflammation [78]. This inflammatory state increases vascular permeability and can activate leukocytes, leading to capillary blockages and local retinal ischemia [79]. Elevated concentrations of pro-inflammatory cytokines and chemokines, encompassing various inflammatory mediators, have been detected in ocular samples from DR patients [80], and upregulated pro-inflammatory cytokines may directly or indirectly induce angiogenesis [81,82]. Research has shown that high-glucose-induced inflammation in DR can be alleviated by either activating [83,84,85] or inhibiting [86] specific signaling pathways.

### 4.3. Advanced Glycation End Products (AGEs)

Persistent hyperglycemia triggers the initiation of non-enzymatic glycation in macromolecules, encompassing proteins, consequently leading to an increase in AGEs [71]. AGEs can increase the levels of cell adhesion molecules within retinal endothelial cells, resulting in capillary occlusion [87]. They may also induce apoptosis of retinal pericytes through oxidative stress mechanisms [88]. The deposition of AGE adducts within the retinal microvascular basement membrane can cause functional impairments, such as perturbation of endothelial junctions and heightened vascular permeability [89], leading to endothelial damage and extravasation of intravascular substances. Additionally, AGEs can cause neuronal abnormalities [87].

### 4.4. Vascular Endothelial Growth Factor (VEGF)

In DR, progressive loss of capillaries leads to retinal hypoxia, inducing the expression of vascular endothelial growth factor (VEGF) [90]. In the context of DR, the progressive decline of capillary density culminates in retinal hypoxia, prompting the upregulation of VEGF. Subsequently, a robust neovascular response is triggered, particularly during the advanced stages of DR [91], characterized by excessive neovascularization [92]. VEGF activates PKC, orchestrating the dismantling of tight junctions, perturbation of the blood–retinal barrier (BRB) and increased permeability of capillaries [93,94].

### 4.5. Protein Kinase C (PKC)

Hyperglycemia causes the buildup of diacylglycerol (DAG), activating various PKC isoforms in the retina [95,96]. PKC promotes the generation of reactive oxygen species (ROS), leading to amplified vascular permeability, upregulation of VEGF expression [97], modifications in blood flow and alterations in enzyme activity [98]. These intricate processes contribute to retinal cell apoptosis, the emergence of capillaries devoid of cellular components and the disturbance of the functionality of the BRB [99].

### 4.6. Polyol Pathway

Hyperglycemia triggers the polyol pathway activation across diverse cellular populations, leading to excessive aldose reductase activity [75]. Consequently, retinal endothelial cells, pericyte cells and other retinal cell types endure detrimental effects mediated by oxidative stress [100], increased cellular osmotic pressure and AGEs formation [101]. It can also lead to abnormalities in neuroglia and neurons [102].

It is important to note that hyperglycemia does not singularly cause retinal damage in DR through only one of the aforementioned pathways. Instead, these pathways interact and influence each other, collectively contributing to vision loss in DR.

## 5. Dietary Modification Strategies

Currently, the main therapeutic approaches for DR include laser photocoagulation, intravitreal administration of pharmacological agents targeting VEGF, intravitreal injection of corticosteroids and vitrectomy [19,24]. However, these approaches primarily target the advanced phases of DR and can only partially alleviate the extent of visual decline; they cannot reverse vision damage. Additionally, these treatments are associated with high costs, multiple side effects and poor patient compliance. Therefore, early prevention is the most effective, cost-effective and beneficial primary choice for delaying vision loss in DR. Among the early prevention methods, dietary modification is one of the simplest and easiest to adhere to.

Since hyperglycemia has a significant implication in the pathogenesis and advancement of DR, dietary modification should focus on reducing sugar consumption. According to guidelines established by The World Health Organization (WHO), limiting the consumption of free sugars to less than 10% of the overall energy intake is recommended. Ideally, it can be controlled below 5% when feasible. Although slight variations exist among different countries and age groups, it is advised to minimize sugar consumption as much as possible. It is imperative to acknowledge that free sugars do not encompass naturally occurring sugars found in fresh fruits, vegetables and milk. Authentic free sugars exclusively include monosaccharides and disaccharides introduced through diverse food manufacturing procedures, commonly recognized as added sugars. Naturally existing sugars in honey and fruit juice also fall under the classification of free sugars [103]. The medical research institute advocates for a maximum threshold of 25% of daily caloric consumption as the recommended limit for added sugar intake [104].

### 5.1. Fruits and Vegetables

Fruits and vegetables hold significant importance as sources of a diverse array of nutrients and dietary fibers [105]. Research indicates that increasing the consumption of fruits and vegetables can effectively diminish the likelihood of developing DR [106,107] and provide protective effects against DR [108]. Fruits and leafy green vegetables contribute to delaying DR progression and mitigating visual impairment [23]. To promote better blood glucose control and mitigate inflammation in sufferers, it is recommended to consume fruits and vegetables that abound in flavonoids, such as leafy greens, fruits and berries [109,110]. These can protect against the demise of retinal ganglion cells (RGCs) triggered by oxidative stress [111].

### 5.2. Fish

Polyunsaturated fatty acids (PUFAs) can protect the vision of patients with DR [112], and reduce the severity of the DR condition [113]. The retina contains a significant concentration of long-chain ω-3 polyunsaturated fatty acids (LCω3PUFAs) that showcase notable anti-inflammatory and antiangiogenic properties. A daily intake of at least 500 mg of dietary LCω3PUFAs can decrease the likelihood of visual impairment in patients of DR [114,115,116,117,118]. Fish, an excellent source of omega-3 polyunsaturated fatty acids, can reduce the formation of pathological blood vessels [119,120]. Increasing fish consumption can slow down the progression of DR [121,122], thereby reducing the probability of developing severe manifestations of the condition [123].

### 5.3. Vitamins

#### 5.3.1. Vitamin A

The administration of vitamin A has been observed to exhibit protective properties against retinal damage induced by hyperglycemia and contribute to the delay in retinal neovascularization formation [124]. Serum levels of vitamin A are correlated with DR [125], emphasizing its potential relevance in assessing the condition.

#### 5.3.2. Vitamin B

Vitamin B3 is characterized by its existence in two distinct forms, namely niacin and niacinamide, each exhibiting unique molecular structures and biological functions. Niacinamide can reduce oxidative deoxyribonucleic acid (DNA) damage, promote DNA repair and alleviate retinal neurodegeneration in diabetic patients [126]. However, high-dose niacin may increase the risk of insulin resistance [127]. Substantial dietary consumption of vitamin B6 has effectively reduced the likelihood of developing DR [128]. Diminished levels of vitamin B12 in the serum have been correlated with an augmented susceptibility to DR [129], with some indications suggesting its potential role as a distinct risk factor contributing to the suffering condition [130].

Additionally, vitamins B1, B7 and B9 have also exhibited significant therapeutic potential in addressing retinal lesions associated with diabetes, underscoring their significant role in managing these pathological conditions [131,132].

#### 5.3.3. Vitamin D

Vitamin D has properties such as lowering blood glucose, antioxidation, anti-inflammation, antiangiogenesis and neuroprotection [133,134,135,136]. There is a compelling correlation between serum vitamin D levels and the occurrence or seriousness of DR [137,138,139,140,141,142,143,144,145]. A prospective study found that maintaining adequate optimal levels of vitamin D in the bloodstream can prevent the deleterious impact of diabetes on the intricately interconnected microvasculature, including DR [146]. The association between serum levels of 25-hydroxyvitamin D and DR was supported by pertinent studies [147,148,149]. When the concentration of serum 25-hydroxyvitamin D falls below the threshold of 15.57 ng/mL, the risk of visual impairment doubles in DR patients [150]. 1,25-dihydroxyvitamin D₃ may serve as a potential protective role in retina by regulating inflammatory responses [151], and possesses inhibitory prowess against the activity of retinal VEGF and transforming growth factor [152]. Fatty fish and fish oil, notably salmon and sardines, are designated as prominent dietary reservoirs of vitamin D [133].

#### 5.3.4. Vitamin E

Elevated blood glucose levels have been implicated as a pivotal contributor in the escalation of oxidative stress, and vitamin E can prevent lipid peroxidation, improve oxidative stress [153] and reduce the severity of diabetes-related complications [154]. Additionally, vitamin E can significantly decrease retinal capillary basement membrane thickness, protecting against retinal damage associated with DR caused by hyperglycemia-induced oxidative stress [155,156].

### 5.4. Non-Vitamin-A Carotenoids

Carotenoids, which are indigenous antioxidants, exhibit both anti-inflammatory and antioxidant qualities. They can reduce inflammation and oxidative stress caused by high blood glucose. Their remarkable impact involves mitigating the inflammatory response and oxidative burden arising from hyperglycemia, thereby slowing down the occurrence and development of DR [157,158]. The main carotenoids include lutein, zeaxanthin, astaxanthin and lycopene.

The macula, a region in the retina, showcases a profound abundance of lutein and zeaxanthin [159]. These two nutrients, well recognized as the macular pigment, possess antioxidant, anti-inflammatory, antiangiogenic, neuroprotective and blue-light-filtering properties in the eyes [160,161,162,163,164,165,166]. They can significantly improve retinal vascular changes caused by high blood glucose [167] and enhance macular function [168], thereby protecting and alleviating DR [169,170,171,172,173]. Carotenoid dietary intake is significantly reduced in DR patients [174]. Since the human body cannot synthesize carotenoids, it is recommended that DR patients obtain them from food sources. Lutein predominantly resides within verdant leafy vegetables such as broccoli, spinach and lettuce [175]. Zeaxanthin is mainly found in corn and corn products [175], while astaxanthin is primarily found in seafood and algae [176,177].

### 5.5. Flavonoids

Flavonoids are the preeminent polyphenols ubiquitously interspersed within our nutritional diet and possess anti-inflammatory and antioxidant properties. They can alleviate hyperglycemia, inhibit oxidative stress and inflammation processes [178], regulate carbohydrate and fat metabolism [179] and slow down visual loss in DR [180,181].

Anthocyanins are a class of flavonoids and essential natural bioactive pigments [182]. They are mainly found in berries and cherries [183], such as wild blueberries, cranberries, raspberry seeds and strawberries [184]. Cyanidin-3-O-glucoside (C3G) is an anthocyanin type with strong antioxidant and anti-inflammatory effects. It exhibits a therapeutic intervention for ameliorating inflammation triggered by elevated glucose levels, along with mitigating angiogenesis in DR [185]. Blueberry anthocyanins can enhance the integrity of the BRB compromised by oxidative stress [186], adverse consequences inflicted by oxidative stress and inflammation on retinal tissue homeostasis [187], and inhibit the progression of DR [188]. Blueberry anthocyanin extract protects the capillaries of the retina from elevated-glucose-level-induced damage through antioxidant and anti-inflammatory mechanisms [189].

Apart from its antioxidant properties that mitigate oxidative stress, the flavonoid naringenin and its derivatives exhibit neurotrophic effects, reducing neuronal vascular damage associated with DR [190]. Genistein, predominantly found in leguminous foods such as broccoli and cilantro, manifests anti-inflammatory properties and suppresses neovascularization in ocular tissues [191].

### 5.6. Dietary Fiber

Dietary fiber manifests an inhibitory impact on the kinetics of monosaccharides and fatty acid digestion and absorption, impeding their prompt assimilation, thereby reducing calorie absorption [192]. The utilization of this therapeutic approach demonstrated a reduction in the likelihood of DR [193] and has a protective effect against existing cases [108]. Investigations have elucidated that individuals with low-dietary-fiber intake have a higher risk of vision-threatening DR [194].

### 5.7. Other Nutrients

Whole grains are rich in soluble and insoluble fiber, which can lower blood glucose levels, ameliorate blood lipid levels and optimize gut microbiota [195]. The consistent incorporation of cheese and whole wheat bread into the diet has been found to correlate with a reduced likelihood of DR advancement [196]. Goji berries can increase the concentrations of lutein and zeaxanthin, ameliorating high-glucose-induced microstructure and physiological damage in the retina [197]. Furthermore, it confers protective effects on the retina [198]. Coconut water can lower blood sugar levels and mitigate DR damage [199].

## 6. Dietary Recommendations for DR Patients—The Mediterranean Diet (MedDiet)

The Mediterranean diet (MedDiet), renowned for its global recognition, originates from the Mediterranean coast regions. This dietary pattern is distinguished by its emphasis on the abundant consumption of plant-derived foods, including fruits, vegetables, legumes, grains and nuts, preferably fresh and minimally processed. It also includes a significant intake of olive oil as the foremost contributor of fat, desired consumption of dairy products like cheese and yogurt, prudent consumption of fish and poultry and moderate alcohol consumption [200,201,202] (Figure 3).

MedDiet is rich in low glycemic foods, vitamins, minerals, antioxidants, fiber, monounsaturated fatty acids (MUFAs), PUFAs and probiotics. Research has evidenced the ability of this intervention to result in reduced levels of HbA1c, concentrations of postprandial blood glucose, oxidative stress and inflammation, and improve lipid profiles and gut immune function [201,203]. Numerous studies have demonstrated that adhering to MedDiet helps to reduce the morbidity of DR and prevents vision loss [204,205,206,207,208].

## 7. Other Lifestyle Recommendations

Hypertension is recognized as an additional contributing factor to the incidence of DR, and reducing elevated blood pressure levels confers valuable preventive benefits against DR [209]. The Dietary Approaches to Stop Hypertension (DASH) diet has been linked to the management of hypertension and shares similarities with MedDiet. It also possesses anti-inflammatory and antioxidant properties, which can improve blood glucose control [210]. Calorie restriction and intermittent fasting have demonstrated potency in reducing blood glucose levels in the blood circulation [211]. Maintaining a low-calorie and low-sodium intake is also beneficial for DR [212]. Hyperglycemia, hypertension and dyslipidemia represent a triad of modifiable primary risk factors for severe vision loss in sufferers [213]. Achieving optimal glycemic control, blood pressure and lipid profiles management can reduce diabetes-related vision impairment [214,215].

Furthermore, engaging in appropriate physical exercises, such as aerobic activities, strength training and yoga, can mitigate the likelihood of DR occurrence and progression [216,217].

## 8. Conclusions

Diabetes manifests as the foremost prevailing metabolic disorder, with diabetic retinopathy emerging as its predominant and consequential complication. Additionally, this ocular condition serves as the primary contributor to avoidable vision loss among the working-age cohort. The occurrence and advancement of DR are influenced by a variety of risk elements, including hyperglycemia, hypertension, dyslipidemia and duration of diabetes. Notably, hyperglycemia assumes a pivotal role as the primary driver behind the occurrence and progression of DR. Therefore, it is crucial to gain a comprehensive understanding of the underlying disease mechanisms through which HSDs intricately contribute to the occurrence and progression of DR. Given the inconspicuous presentation of clinical indicators amidst the nascent stages of microvascular and neuronal compromise in DR, it is challenging to raise awareness among patients. However, once clinical symptoms manifest, vision loss becomes inevitable. Existing treatment options can only partially alleviate the degree of visual impairments but cannot reverse visual damage. Therefore, early intervention is of utmost importance.

Among various early intervention methods, dietary intervention is the most cost-effective, patient-friendly, least harmful and highly beneficial approach. We primarily elucidate the complex pathogenesis of DR and the intricate biochemical mechanisms by which a high-sugar diet leads to irreversible vision loss in DR. This aims to raise awareness about the importance of daily dietary habits in the occurrence and advancement of DR. Moreover, we underscore the pronounced impact of specific nutrients in delaying the development of DR, providing dietary modification strategies for DR patients as references. It is noteworthy that the application of a balanced low-sugar diet is the primary factor in the prevention and therapy of hyperglycemia and its bad consequences, and that patients with DR must integrate the use of necessary anti-hyperglycemic and lipid-lowering medications, such as statins, alongside appropriate dietary adjustments and suitable physical exercise. A singular dietary adjustment alone is unlikely to exert a decisive impact on the progression of DR. Therefore, the adoption of a balanced low-sugar diet complements pharmacological interventions, facilitating enhanced therapeutic efficacy, alongside the indispensable inclusion of daily physical activity in routine life. We recommend that DR patients minimize their daily sugar intake while controlling the three major high-risk factors of glycemic, blood pressure and lipid levels to mitigate the potential hazards of vision loss as much as possible.

## Figures and Tables

**Figure 1 nutrients-16-01393-f001:**
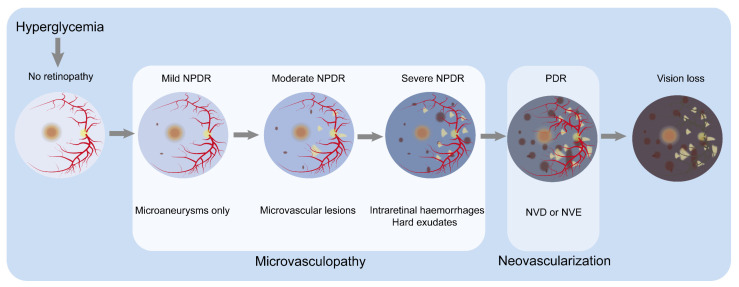
Classification of DR by severity and the major clinicopathological features associated with different stages. In no retinopathy, the retina shows no microvascular abnormalities. Hyperglycemia damages the normal retina, resulting in mild NPDR, characterized by microaneurysms. In moderate NPDR, microaneurysms and other microvascular abnormalities are observed. Severe NPDR is basically characterized by one or a combination of the following: (1) more than 20 retinal hemorrhages; (2) venous beading; (3) retinal microvascular abnormalities but not meeting the criteria for PDR. The progression to the PDR stage is marked by the appearance of NVD or NVE, along with preretinal or vitreous hemorrhage. Microvasculopathy is the main characteristic of the NPDR stage, while neovascularization is the main characteristic of the PDR stage. As the condition worsens, patients experience visual loss. Abbreviations: neovascularization of optic disc, NVD; neovascularization of elsewhere, NVE; non-proliferative diabetic retinopathy, NPDR; proliferative diabetic retinopathy, PDR.

**Figure 2 nutrients-16-01393-f002:**
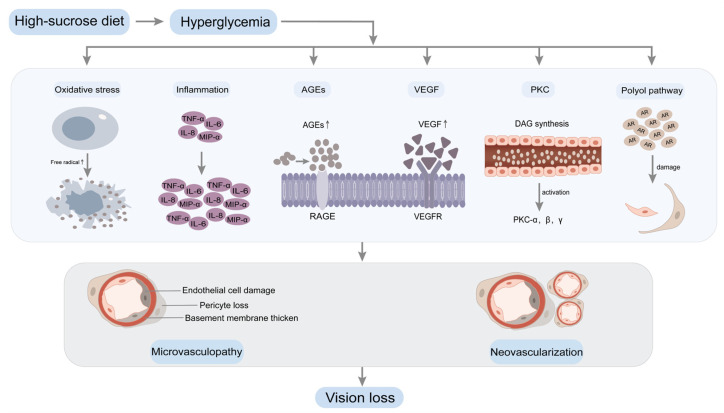
Hyperglycemia induces biochemical changes in DR. A high-sucrose diet leads to increased blood glucose levels, resulting in hyperglycemia. Prolonged and sustained high blood glucose levels contribute to retinal microvasculopathy through multiple pathways, including oxidative stress, inflammation, AGEs, VEGF, PKC and the polyol pathway. Significant apoptotic events within the retinal capillary cells instigate retinal hypoxia and subsequent neovascularization. The proliferation of abundant neovascularization engenders the development of fibrous tissue covering, leading to tractional retinal detachment. Ultimately, this culminates in vision loss. Abbreviations: advanced glycation end products, AGEs; receptor for advanced glycation end products, RAGE; vascular endothelial growth factor, VEGF; vascular endothelial growth factor receptor, VEGFR; protein kinase C, PKC; diacylglycerol, DAG; aldose reductase, AR; tumor necrosis factor-α, TNF-α; macrophage inflammatory protein-1α, MIP-α; interleukin-6, IL-6; interleukin-8, IL-8.

**Figure 3 nutrients-16-01393-f003:**
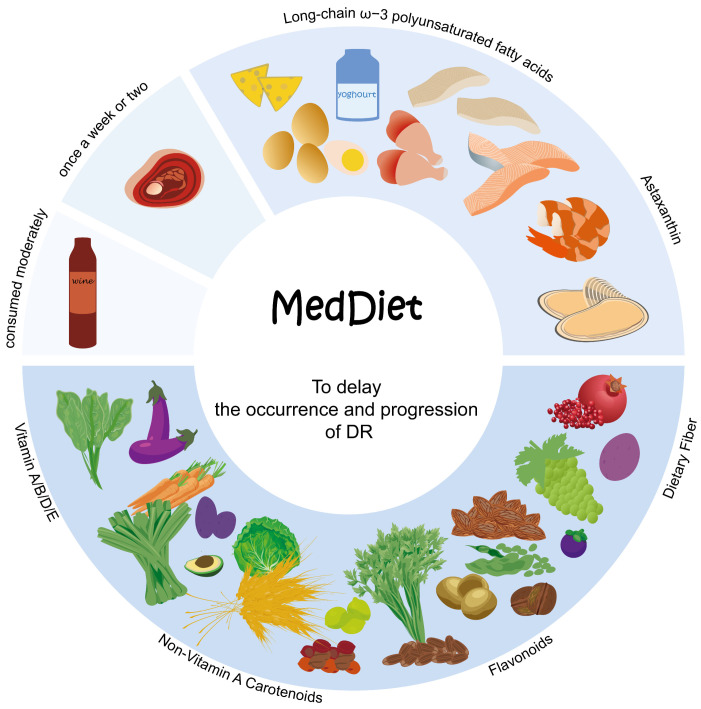
The distinguishing attributes of MedDiet. First, an adequate intake of a huge variety of plant-based foods that are minimally processed, seasonally fresh and locally grown. This includes fruits, vegetables, minimally refined grains, beans and nuts, as they provide a rich array of nutrients such as vitamins, non-vitamin-A carotenoids, flavonoids and dietary fiber. Fresh fruits are incorporated as a regular dessert option. Second, there is a moderate intake of local products derived from milk, primarily yogurt, alongside fish and seafood, to obtain nutrients like LCω3PUFAs and astaxanthin. Third, the consumption of red meat, including processed variants, is limited to once a week or every two weeks, while allowing for moderate wine consumption with meals. MedDiet offers a variety of nutrients that can help to lower blood glucose levels, reduce oxidative stress and mitigate inflammation, thereby exerting a significant impact on retarding the occurrence and progression of DR.

## Data Availability

No new data were created or analyzed in this study. Data sharing is not applicable to this article.

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
