# Peer review of "Effect of High-Sucrose Diet on the Occurrence and Progression of Diabetic Retinopathy and Dietary Modification Strategies"

_nutrients, 2024, doi:10.3390/nu16091393_

Round 1

Reviewer 1 Report

Comments and Suggestions for Authors

This is a very interesting review article that is worthy considering after a minor revision. Only small changes should be made:

1) The Authors could emphasize even stronger that an application of a balanced low-sugar diet is the most important factor in prevention and therapy of hyperglycemia and its bad consequences. The proper diet must be combined with a daily physical activity. An application of anti-hyperglycemic drugs is also of importance, but it must be pointed out that the drugs will not act effectively unless they are combined with a proper diet and a physical activity.

2) The Authors may consider a role of other well-known flavonoids, such as, for example, naringenin and its derivatives, or an isoflavone genistein. Other naturally occurring compounds for a consideration may be statins, which are applied in a treatment of hypercholesterolemia. A reduction of blood cholesterol level may be helpfull in a treatment of blood hypertension, which is another risk factor for the DR.

Minor points:

1) The Authors should provide a list of abbreviations they used in the text.

2) Figure 2 looks too complicated. A simpler version should be prepared.

3) The Authors should read throroughly the text in order to check the English they used.

Reviewer 2 Report

Comments and Suggestions for Authors

MANUSCRIPT: 2966060

TITLE: Effect of High Sucrose Diet on the Occurrence and Progression of Diabetic Retinopathy and Dietary Modification Strategies

The manuscript 2966060 “Effect of High Sucrose Diet on the Occurrence and Progression of Diabetic Retinopathy and Dietary Modification Strategies” presents a review of the literature.

The manuscript presented is well structured.

The review is clearly written, well systematized and comprehensive for the topic, and the literature cited is adequate and most of the papers cited, around 40% of the 206 references cited are from the last five years, which demonstrates the relevance and interest in the topic benefits of the Mediterranean diet.

Similar reviews are not known, and it is of much interest to the scientific community.

The conclusions are consistent and in accordance with the listed citations.

I congratulate the authors for this systematic review of high interest to the scientific community where they present a review that is very easy to read.

However, I have only one minor question to be clarified and resolved and the manuscript in the current form must be revised according to the comment below: 

1. Figures 1, 2 and 3. Authors must be aware that the figures are not their own. Please note that you cannot use figures already published in works by other authors without permission. Figures taken directly from the literature/references without alteration should include the original reference source in the respective figure caption and should say “Taken from Ref. XX with permission of (publisher)” or “Published with permission of XX publisher” and cite the publication. Modified figures should say “Adopted from Ref. XX” and nothing else is needed. Please check if the figures were obtained from already published works and proceed as described in this point.

Reviewer 3 Report

Comments and Suggestions for Authors

Thank you for the manuscript. It is well-written, but there are some spelling errors throughout the text, such as:

-          In the abstract, it should be: "an excellent outcome."

-          All abbreviations must be explained upon first use. For example, what does "DR" stand for? Please clarify as "diabetic retinopathy (DR)," "AGE," "VEGF," and I suggest including an abbreviation list at the end of the article.

-          Abbreviations should be written out correctly, for instance: "non-proliferative diabetic retinopathy (NPDR);" "proliferative diabetic retinopathy (PDR)."

-          In the sentence "the activation of the PKC pathway," please explain what "PKC" stands for.

-          Similarly, for "formation of AGEs," please provide an explanation for the abbreviation.

-          Instead of "diabetic suffer," it should be "diabetic sufferers."

-          For "for patients with DR, earlier intervention, timely diagnosis and treatment," it should be "treatments."

-          Abbreviations should be expanded, like "long-chain ω-3 polyunsaturated fatty acids (LCω3PUFAs)" and "Deoxyribonucleic Acid (DNA)."

-          There should be a space before every reference in the text.

-          In the sentence "Research has shown that alleviating high glucose-induced inflammation in DR can be achieved by either activating[74-76]or inhibiting[77]specific signaling pathways," it would be better to place the references at the end of the sentence or insert spaces in the text for better readability.

These minor issues should be addressed. The topic is very important. Unfortunately, I did not receive a Turnitin report, so I cannot assess the similarity index. Thank you.

Comments on the Quality of English Language

-
